# Unilateral Mitochondrial–Hemodynamic Coupling and Bilateral Connectivity in the Prefrontal Cortices of Young and Older Healthy Adults

**DOI:** 10.3390/bioengineering10111336

**Published:** 2023-11-20

**Authors:** Claire Sissons, Fiza Saeed, Caroline Carter, Kathy Lee, Kristen Kerr, Sadra Shahdadian, Hanli Liu

**Affiliations:** 1Department of Bioengineering, University of Texas at Arlington, Arlington, TX 76019, USA; 2School of Social Work, University of Texas at Arlington, Arlington, TX 76019, USA

**Keywords:** age difference in prefrontal cortex, mitochondrial and hemodynamic coupling, prefrontal cortical connectivity, resting state functional connectivity, broadband near-infrared spectroscopy

## Abstract

A recent study demonstrated that noninvasive measurements of cortical hemodynamics and metabolism in the resting human prefrontal cortex can facilitate quantitative metrics of unilateral mitochondrial–hemodynamic coupling and bilateral connectivity in infraslow oscillation frequencies in young adults. The infraslow oscillation includes three distinct vasomotions with endogenic (E), neurogenic (N), and myogenic (M) frequency bands. The goal of this study was to prove the hypothesis that there are significant differences between young and older adults in the unilateral coupling (uCOP) and bilateral connectivity (bCON) in the prefrontal cortex. Accordingly, we performed measurements from 24 older adults (67.2 ± 5.9 years of age) using the same two-channel broadband near-infrared spectroscopy (bbNIRS) setup and resting-state experimental protocol as those in the recent study. After quantification of uCOP and bCON in three E/N/M frequencies and statistical analysis, we demonstrated that older adults had significantly weaker bilateral hemodynamic connectivity but significantly stronger bilateral metabolic connectivity than young adults in the M band. Furthermore, older adults exhibited significantly stronger unilateral coupling on both prefrontal sides in all E/N/M bands, particularly with a very large effect size in the M band (>1.9). These age-related results clearly support our hypothesis and were well interpreted following neurophysiological principles. The key finding of this paper is that the neurophysiological metrics of uCOP and bCON are highly associated with age and may have the potential to become meaningful features for human brain health and be translatable for future clinical applications, such as the early detection of Alzheimer’s disease.

## 1. Introduction

The enigmatic puzzle of brain function in young and older adults has intrigued scientists for years, including the prefrontal cortex, which is closely associated with human cognition as evidenced by many studies [1,2,3,4,5]. As aging progresses, a complex interplay of neurophysiological changes, such as mitochondrial and vascular alterations, shapes the performance of the human brain in young and older adults, unveiling cognitive decline and its impact on executive mechanisms [6,7]. Recently our research team developed a new noninvasive measurement and quantification method that permits objective estimates of unilateral mitochondrial–hemodynamic coupling and their bilateral connectivity in the human frontal cortex [8]. This study focused on the impact of aging on the unilateral coupling and bilateral connectivity of mitochondrial and vascular/hemodynamic alterations in the resting prefrontal cortex in two distinct age groups. We aimed to shed light on how aging influences local and bilateral coherence and thus the connectivity of neurophysiological activity in the prefrontal regions. Understanding the age-related shifts in neurophysiological mechanisms is crucial because they may impact various aspects of daily life, including decision-making, problem-solving, and attentional control [9].

### 1.1. Unilateral Metabolic–Hemodynamic Coupling and Bilateral Connectivity

Because of its high-level neuronal activity and cerebral metabolism, the human brain generally consumes a large amount of oxygen and glucose, which is continuously supported by an abundant oxygenated blood supply [8]. Cytochrome c oxidase (CCO) is a mitochondrial enzyme that consumes oxygen to produce adenosine triphosphate (ATP). The continuous and active synthesis of ATP by CCO provides intracellular energy to neurons and is essential for the normal operation of the human brain [10,11]. Thus, a close correlation or coherence must exist between local mitochondrial (or metabolic) and neurovascular (or hemodynamic) coupling, which is termed metabolic–hemodynamic coupling hereafter. Specifically, this study investigated unilateral coupling (uCOP) on each side of the human prefrontal cortex, namely, uCOP_left_ and uCOP_right_. These two quantifiers reflect the balance between local neuronal metabolism (i.e., the redox state of CCO) and oxygenated hemodynamics (i.e., oxygenated hemoglobin concentration, HbO) on each lateral side of the prefrontal cortex [8]. In the meantime, we also calculated the bilateral coherence of HbO (or CCO) signals of the prefrontal cortex, which is termed bilateral hemodynamic (or metabolic) connectivity, bCON_HbO_ (bCON_CCO_), of the resting prefrontal cortex. A larger coherence value indicates stronger bilateral connectivity [12].

### 1.2. Three Infraslow Oscillation Bands in Cerebral CCO and HbO Signals

Vasomotion is a key contributor to metabolic and hemodynamic oscillations and has been explored in studies focusing on cerebral metabolic activity [6,7]. The relaxation and contraction patterns of the blood vessel walls have been identified as the driving forces behind the infraslow oscillations observed in cerebral hemodynamic signals [13,14,15,16]. Researchers have discovered three intrinsic rhythm bands within cerebral hemodynamic signals that align with the specific physiological and/or biochemical activities of the vascular wall layers [8,10,17]. The three frequency bands were classified as endogenic (E) (0.005–0.02 Hz), neurogenic (N) (0.02–0.04 Hz), and myogenic (M) (0.04–0.2 Hz) rhythms, as shown in Figure 1a [8,18,19]. The E band is caused by endothelium-derived vasomotor activities, which are mediated by mural cells. The N band involves the modulation of vessel dilation and contraction cycles by the oscillation of vasoactive ions and neurotransmitters released from neurons. Lastly, rhythmic M activity is a result of the relaxation and contraction of smooth muscle cells within the vascular walls. Figure 1a illustrates the anatomical structures and respective rhythm frequencies of the three infraslow oscillation components at a local cerebral site [20]. Figure 1b,c further graphically explain the bilateral prefrontal connectivity and unilateral coupling derived from the HbO and CCO signals, respectively. The mathematical principle and operation to derive the three E/N/M bands are shown schematically in Appendix A.

Over time, the blood vessel walls become weaker or stiffer, causing a reduction in hemodynamic rhythms in the human brain. For example, vasomotion malfunction has been observed in older adults and in patients with different diseases, such as atherosclerosis [22], cardiovascular disease [23], and Alzheimer’s disease [24]. Thus, it may be beneficial to quantify and characterize cerebral metabolism in the infraslow oscillation range, which may provide better insight into neurophysiological mechanisms and discover features that differ between healthy young and older adults for the early diagnosis of Alzheimer’s disease and other brain disorders.

### 1.3. Aim of This Study

Numerous studies have reported differences in brain function in response to different cognitive tasks in the prefrontal cortices of young and older individuals [6,25,26,27]. For example, healthy aging brains exhibited increases in bilateral activation of the frontal and prefrontal cortices when performing the tasks that stimulated only unilaterally in young adults [6]. Studies examining the impact of age on task performance have consistently shown that young adults perform better on measures of frontal lobe function than older adults, while both groups perform equally well on tasks that do not primarily rely on the frontal lobe [25]. In contrast, this study aimed to identify and quantify the differences between young and older adults in a newly defined set of neurophysiological metrics (i.e., uCOP_left_ and uCOP_right_, bCON_HbO_, and bCON_CCO_) in the resting prefrontal cortex.

The four new neurophysiological metrics can be derived from noninvasive measurements of two-channel broadband near-infrared spectroscopy (bbNIRS) obtained from the forehead of human participants [8,19,21]. This portable device employs near-infrared light to non-invasively detect changes in chromophore concentrations, encompassing oxidized cytochrome c oxidase (Δ[CCO]) and oxygenated hemoglobin (Δ[HbO]) [28,29]. In this study, we first quantified these four metrics in older adults (*n* = 24) at all three infraslow oscillation (E/N/M) frequencies and then compared them with the respective metrics in young adults (*n* = 26), followed by examining possible gender differences within each age group. The aim of this study was to prove our hypothesis that there are significant differences in unilateral metabolic–hemodynamic coupling and bilateral connectivity in the prefrontal cortex between young and older adults. Such new findings may be potentially meaningful for clinical applications, since impairment in unilateral/local coupling or the bilateral connectivity of prefrontal infraslow oscillations may signify neurological disorders [1,5,8].

## 2. Materials and Methods

### 2.1. Participants

This study employed a between-group experimental design with two age groups. Twenty-six healthy (fourteen males and twelve females; mean ± SD age = 22.4 ± 2.3 years) were recruited from the university community and reported in a recent publication [1]. For this study, each young participant had 5 visits, separated by at least 7 days. They were screened using the same inclusion/exclusion criteria as those used in previous studies [10,30]. In addition, we recruited 24 older adults (4 males and 20 females; mean ± SD age = 67.2 ± 5.9 years) from the Dallas–Fort Worth community and were screened with the following inclusion criteria: (1) aged 55 years or older, (2) not experiencing cognitive impairment or decline, and (3) being able to travel without assistance to the research laboratory on the UT Arlington campus. Those who had brain injuries or surgeries within the past year or could not provide consent of their own were excluded from the healthy subject category. The experimental procedures (see below) were approved by the Institutional Review Board of the University of Texas at Arlington. All measurements were conducted after obtaining informed consent. The participants were compensated for their time.

### 2.2. Experiment Protocol and Setup

Both age groups underwent the same experimental protocol with the same experimental setup, which was reported in detail in ref. [8]. For the reader’s convenience, we briefly summarize the experimental protocol and setup as follows. Each older adult was asked to complete a demographic questionnaire and sign a consent form, followed by sitting comfortably on a sofa chair. As shown in Figure 2a,b, a two-channel bbNIRS headset was positioned bilaterally on the participants’ foreheads, along with a 19-channel EEG cap. After the completion of bbNIRS + EEG setup, each measurement took 14-min readings while the participant was in a resting state with their eyes open and closed for 7 min each (Figure 2c). After completion of the 14-min data collection, only two-channel bbNIRS data were used for further data processing for this study. 

### 2.3. Broadband Near-Infrared Spectroscopy and Its Measurements

As shown in Figure 2b and refs. [8,21], the bbNIRS system employed in this study was designed with two channels to collect data concurrently from each lateral forehead of the participant. The two light sources used were halogen lamps (OSL2IR, Thorlabs, Inc., Newton, NJ, USA) emitting broadband white light. Two CCD array spectrometers (QEPRO; Ocean Optics Inc., Orlando, FL, USA) were used for spectroscopic detection. An integration time of 1.5 s (i.e., a sampling rate of 0.67 Hz) was set to balance the temporal resolution and adequate signal strength. Two sets of optical fiber probes were connected to a laptop computer that controlled the data acquisition, displayed the results, and stored the data for the offline process. Calibration of the two spectrometers was performed using an ink-intralipid phantom, demonstrating identical spectral quantifications from both channels. 

To ensure proper placement, optical fibers connected the lamps and detectors to a 3D-printed headband, as seen in Figure 2a,b. The headband was secured to each participant’s forehead using Velcro straps and medical tapes applied to the probe–skin interface to stabilize the probe without causing discomfort to the participants. In addition, this stabilization greatly reduces motion artefacts. This headband was specifically designed with divots to accommodate the EEG prefrontal electrodes while securing the placement of the bbNIRS system on the subject’s forehead. The source and detector separation for each channel was 3 cm. This setup enables the simultaneous measurement of optical spectral alterations in both the left and right foreheads of healthy participants in the resting state [8]. 

In the meantime, a compact electroencephalogram (EEG) device with a dry, blue-tooth controlled, 19-channel headset (Quick-20, CGX Systems, San Diego, CA, USA) was employed for concurrent dual-mode measurements. However, the focus of this study was on the results only from the bbNIRS of the two age groups, leaving the investigation of EEG to a future study.

### 2.4. Data Analysis

The algorithm developed to analyze two-channel bbNIRS data for the quantification of bilateral connectivity of ΔHbO and ΔCCO, as well as unilateral mitochondrial–hemodynamic coupling, of the human forehead was recently introduced [1] and applied to the investigation of prefrontal responses to noninvasive light stimulation [19,21]. The detailed data analysis can be summarized in five steps, as shown in Figure 2d. While complete derivation and explanation of the algorithm can be found in refs. [1,2], we briefly describe the five steps below with a graphical illustration (Figure 3) for the convenience of the readers.

Step 1: Raw bbNIRS data collection from both older and young adult groups

For both older adult and young adult experiments, the data recorded by both spectrometers were a set of optical spectra at different times (*t*), as expressed *I*(*t*, *λ*). A relative optical density spectrum, ΔOD(*t*, *λ*), can be defined and calculated at each wavelength *λ* as [10]:(1)$$\Delta OD(t,\lambda )=\log_{10}[\frac{I_{0}(t=0,\;\lambda )}{I(t,\lambda )}],$$
where *I*_0_(*t =* 0, *λ*) is the baseline spectrum at time *t* = 0 or an average of several initial baseline spectral readings and *I*(*t*, *λ*) represents time-varying spectra acquired at each time point throughout the entire experiment. 

Step 2: Conversion of ΔOD(*t*, *λ*) to Δ[HbO](*t*, *λ*) and Δ[CCO](*t*, *λ*) over the 7-min resting state

With a sample rate of 0.67 Hz, 7-min bbNIRS data collection provided a set of 280 spectra for either the eyes-open or eyes-closed session. The recorded spectrum was in the wavelength range of 740 to 1100 nm, but a spectral band of 780 to 900 nm was sufficient for our needs [31]. According to the modified Beer–Lambert law and diffusion theory [32], we converted ΔOD(*t*, *λ*) to Δ[HbO](*t*, *λ*) and Δ[CCO](*t*, *λ*) [33,34,35] for each time point over the 7-min measurement window for both lateral sides of the measurements [8,10,19]. Appendix A provides theoretical/mathematical derivations in detail for this step. 

Step 3: Spectral analysis of Δ[HbO](*t*, *λ*) and Δ[CCO](*t*, *λ*)

To perform spectral analysis for time series of Δ[HbO] and Δ[CCO], we used the multi-taper method (MTM) [8,36,37]. This method facilitates frequency spectra for both Δ[HbO] and Δ[CCO] with relatively high spectral resolution and low noise using Slepian sequences to taper time series in the time domain followed by the fast Fourier transform. Specifically, several functions available in the FieldTrip toolbox (including “ft_freqanalysis”) were performed for MTM operation on the MATLAB platform [38,39]. The decomposed amplitude and phase were achieved as a complex number that was further used in the coherence quantification (see Step 4). See Appendix A for a graphical explanation for the function of “ft_freqanalysis.” 

Step 4: Hemodynamic and metabolic connectivity/coupling quantification [8,21]

Connectivity analysis, in principle, estimates the level by which two time series oscillate synchronously. One of the widely used connectivity measures is coherence, a phase-based frequency-domain analysis that is quantified as a normalized value between 0 and 1. Mathematically, the representation of coherence (*coh_xy_*) between two time series, x and y, for a specific frequency of *ω* is: (2)$${coh}_{xy}\left(\omega \right)=\frac{\left\lceil S_{xy}(\omega )\right\rceil }{\sqrt{S_{xx}\left(\omega \right)S_{yy}(\omega )}}$$
where *S_xx_* and *S_yy_* are the power estimates of signals *x* and *y*, and *S_xy_* is the averaged cross spectral density of the two data series [40]. These terms are calculated using the complex values obtained from the MTM method [37,41] (see Step 3 above). The flow chart shown in Appendix A also illustrates the function of “ft_connectivityanalysis” graphically for an easy understanding of the calculation.

In this study, we utilized several functions in MATLAB (including “ft_connectivityanalysis”) available in the FieldTrip toolbox to perform coherence operations. Specifically, we calculated coherence values for the following four pairs of measured Δ[HbO] and Δ[CCO] signals: (1) bilateral hemodynamic connectivity between Δ[HbO]_right_ and Δ[HbO]_left_ (i.e., bCON_HbO_), (2) bilateral metabolic connectivity between Δ[CCO]_right_ and Δ[CCO]_left_ (bCON_CCO_), (3) unilateral hemodynamic–metabolic coupling on the ipsilateral side between Δ[HbO]_right_ and Δ[CCO]_right_ (uCOP_right_), and (4) unilateral hemodynamic–metabolic coupling on the contralateral side between Δ[HbO]_left_ and Δ[CCO]_left_ (uCOP_left_). These calculations were performed separately for the three frequency bands (E/N/M). 

Step 5: Statistical Analysis

After the aforementioned parameters at each E/N/M band were quantified for older adults, we performed two-sample t-tests between the older and young adults to determine whether each of the bCON and uCOP parameters was age-dependent. The significance level was set at *p* < 0.05. All respective values for the young adult group were based on the results reported in ref. [8]. When a significant difference between the two groups was obtained, we further calculated Cohen’s d to assess the effect size of statistical significance. Accordingly, 0.2 < d < 0.5, 0.5 < d < 0.8, 0.8 < d < 1.3, and d > 1.3 are considered small, medium, large, and very large effect sizes, respectively. 

Furthermore, within each age group, we performed two-sample t-tests between male and female participants to examine the gender difference for each of the quantified connectivity and coupling parameters. Within the older adult group, we performed paired *t*-tests to examine significant differences (i) between eyes-open and eyes-closed conditions and (ii) between unilateral metabolic–hemodynamic couplings in the two lateral prefrontal cortices.

## 3. Results

The results are reported below in four subsections, the first two of which are for between-group comparisons and the last two are for within-group comparisons. To be comparable with the young adult group, only the eyes-closed measurement results of older adults were used for between-group comparisons.

### 3.1. Between-Group Comparisons of Bilateral Prefrontal Connectivity (bCON)

The bilateral prefrontal connectivity of Δ[HbO] and Δ[CCO] (i.e., bCON_HbO_ and bCON_CCO_) from older (*n* = 24) and young adults (*n* = 26 [8]) in the eyes-closed resting state is plotted in Figure 4a and Figure 4b, respectively, for all three E/N/M bands. Figure 4a shows that young adults had significantly stronger connectivity of the bilateral hemodynamics in the prefrontal cortex only in the M band. This significant difference had a very large effect size of 2.35, as listed in Table 1a. In contrast, Figure 4b illustrates that older adults had much stronger metabolic bilateral connectivity, bCON_CCO_, than younger adults in the M band, with a very large effect size of 2.99, as listed in Table 1b.

Note that the bCON_HbO_ and bCON_CCO_ values for young adults (*n* = 26) were grand averaged over five visits [8], while older adults had only one visit for the measurements (*n* = 24). The outcome of such a difference in the sampling size between the two groups is addressed in the Section 4. 

### 3.2. Between-Group Comparisons of Unilateral Prefrontal Coupling (uCOP)

The same coherence analysis was performed to find the unilateral hemodynamic–mitochondrial coupling on the left and right side of the prefrontal cortex from older adults (*n* = 24) and young adults (*n* = 26 [8]) in the eyes-closed resting state. The results are plotted in Figure 5a,b for all three E/N/M bands. As shown in Figure 5, older adults exhibited significantly larger coherence between ∆[HbO] and ∆[CCO] oscillations in all three frequency bands, indicating stronger mitochondrial–hemodynamic couplings on either side of the prefrontal cortex (i.e., uCOP_left_ and uCOP_right_) than in younger adults. In particular, such unilateral coupling on both lateral sides of the older adults was much stronger in the myogenic frequency band than in the young adult group, with very large effect sizes of 1.96 and 2.72 (see Table 2). Unilateral coupling strengths at the endogenic and neurogenic frequency bands in the older adult group were significantly larger with a medium effect size (0.5–0.8) than those in the young adult group (Table 2). 

### 3.3. Comparisons of bCON and uCOP Metrics under Eyes-Open and Eyes-Closed Conditions in Older Adults

Figure 6 below shows comparisons of respective bCON and uCOP metrics taken under 7-min eyes-closed and 7-min eyes-open conditions from older adults in all three E/N/M bands. All older participants were measured under both conditions. Thus, paired t-tests were performed to determine any significant difference in each of the bCON and uCOP metrics between the two conditions. The statistical analysis revealed no significant difference in any of the bCON and uCOP parameters in all E/N/M bands caused by these two conditions.

Furthermore, we also performed paired *t*-tests between the values of uCOP on both the left and right prefrontal cortex for older adults to examine whether there existed any significant difference in metabolic–hemodynamic coupling between the two prefrontal cortices. The statistical analysis showed no significant difference in uCOP between the two prefrontal cortices.

### 3.4. Gender Comparisons of bCON and uCOP Metrics in Young Adults

Besides examining the age effect on resting state bCON and uCOP in the prefrontal cortex of healthy adults, it is also meaningful to inspect the gender effect on those respective metrics within each age group. However, the older adult group had 4 males and 20 females, which created a high imbalance between the sexes, and we had no rationale to study the gender effect. Accordingly, we quantified respective metrics within the young adult group in all three E/N/M bands, as shown in Figure 7. 

## 4. Discussion

The experimental results shown in Section 3 revealed three key findings. First, young adults had significantly stronger bilateral hemodynamic connectivity (bCON_HbO_), as represented by the temporal coherence of resting HbO in the M band compared to older adults (Figure 4), with a very large effect size (=2.35). Second, on the other hand, older adults had significantly stronger bilateral metabolic connectivity (bCON_CCO_) in the M band than young adults (Figure 4b), also with a very large effect size (=2.99). Third, older adults also had significantly stronger unilateral coupling on both prefrontal sides (uCOP_left_ and uCOP_right_) in all three E/N/M bands (Figure 5), with very large effect sizes for the M band (>1.9) and a medium effect size for the E/N bands (0.4 < effect size < 0.8; see Table 2). All of these results clearly proved our hypothesis that there are significant differences in unilateral metabolic–hemodynamic coupling and bilateral connectivity in the prefrontal cortex between young and older adult groups.

### 4.1. Age Effect on Bilateral Hemodynamic Connectivity of the Resting Prefrontal Cortex

Figure 4a demonstrates that both young and older adults had strong bilateral hemodynamic connectivity in all three E/N/M oscillation bands, with bCON_HbO_ larger than 0.7, except for older adults in the M band. It is known that cerebral blood vessel walls in the human brain oscillate in infraslow rhythms in all three E/N/M bands [13,14,15,16]. Thus, this set of results was expected in that the hemodynamic coherence/connectivity between the two lateral prefrontal cortices would be large in both age groups. However, the older adult group exhibited significantly weaker connectivity in the M band than the young adult group. This implies that the synchrony between the smooth muscle cells covering the vascular walls on both lateral sides of the prefrontal cortex is lower in older participants than in younger participants. We speculated that this age effect could be attributed to the aging smooth muscle cells of the cerebral blood vessels that become less elastic and have reduced oscillation frequencies [8]. However, synchronized hemodynamic activity in both E and N bands, mediated by endothelial cells and interneurons, remained unaffected by age. Note that our interpretation of the reduction in bilateral connectivity in the M band in older adults is consistent with a previous connectivity study [42]. According to Li et al. [42], older adults showed an overall decline in both global and local network efficiency compared with young adults. In addition, young adults showed an abundance of brain network hubs in the prefrontal cortex, whereas older adults showed hub shifts to the sensorimotor cortex. 

### 4.2. Age Effect on Bilateral Metabolic Connectivity of the Resting Prefrontal Cortex

We introduced the concept of bilateral metabolic connectivity in the resting prefrontal cortex and quantified it in all three E/N/M bands in young participants in ref. [8]. In this study, we compared these quantities of 24 older adults with those of 26 young adults [8], as presented in Figure 4b. Our observations included (1) that both age groups had much smaller bilateral mitochondrial connectivity values (with bCON_CCO_ being 0.3–0.4) than those of bCON_HbO_ (with it being 0.7–0.8) in all three E/N/M bands and (2) that young adults had significantly weaker bCON_CCO_ values than older adults in the M band with a very large effect size (=2.99). Observation (1) is expected because cerebral mitochondrial activity reflected by the redox state of CCO occurs more locally on each lateral side of the prefrontal cortex and thus is less synchronized between the two sides in all three frequency bands. Observation (2), however, reveals that the prefrontal cortex of young adults operates more independently between the two lateral sides for localized cerebral metabolism with much less bilateral synchrony in the M band than older adults. This may imply that as people age, bilateral prefrontal synchronization of cerebral metabolism increases along with the oscillation of the smooth muscle cells of blood vessels. Thus, bCON_CCO_ in the M band is highly age-dependent and may be an excellent candidate to serve as a marker for age-related brain state conditions.

### 4.3. Age Effect on Unilateral Metabolic–Hemodynamic Coupling of the Resting Prefrontal Cortex

The next metric showing a significant age effect was the unilateral hemodynamic–mitochondrial coupling (uCOP) on the two lateral sides of the forehead. Overall, older adults exhibited significantly stronger unilateral coupling on both lateral sides (uCOP_left_ and uCOP_right_) in all three E/N/M bands (Figure 5), with a medium effect size for the E and N bands (0.4 < effect size < 0.8) and a very large effect size for the M band (>1.9) (see Table 2). Note that the values of uCOP_left_ and uCOP_right_ in older adults were almost twice as large as those in young adults. The neurophysiological measure of uCOP reflects the mitochondrial–vascular or metabolic–hemodynamic interactivity on each individual prefrontal side. Furthermore, the uCOP value signifies direct coupling or the relationship between metabolic demand and blood supply. Accordingly, our results imply that older adults have significantly tighter/closer relationships between local mitochondrial activity (reflected by the redox state of CCO) and oxygenated blood supply (revealed by HbO signals). In other words, the prefrontal cortex in older adults needs to have a faster or better supply-to-demand chain in the cerebral substrate than that in young adults to maintain active and normal prefrontal function in the resting state. This interpretation seems to work for all three frequency bands, while such neurophysiological coupling is strongest with smooth muscles in the M band for the older adult group. 

### 4.4. Signals Measured under Eyes-Open and Eyes-Closed Conditions 

It is known that the resting human brain often exhibits different EEG temporal and spatial signal patterns when the measurement is taken with the eyes open or closed. However, it was unknown whether prefrontal hemodynamic and/or metabolic measurements would differ under eyes-open or eyes-closed conditions. This study clearly indicated that none of the bilateral connectivity metrics (i.e., bCON_HbO_ and bCON_CCO_) or unilateral metabolic–hemodynamic coupling (uCOP_left_ and uCOP_right_) in all three E/N/M bands showed any significant difference under either eyes-open or eyes-closed measurement conditions (Figure 6). This observation experimentally confirms that the prefrontal neurophysiological metrics of a resting brain can be measured with either the eyes open or eyes closed without any significant impact on the results. This finding provides practical and helpful guidance for future experimental designs. 

### 4.5. Gender Difference in Resting Bilateral Connectivity and Unilateral Coupling

Although our results showed age effects on both bCON and uCOP in all three E/N/M bands, we wondered whether these metrics would have a gender effect. Subsequently, for the young adult group, we conducted two-sample t-tests to investigate the potential gender differences for each of the uCOP and bCON metrics. However, no significant evidence was found within the group for most of the metrics, as shown in Figure 7, except for two of the 12 metrics. Specifically, young males showed significantly higher uCOP_right_ and bCON_HbO_ values in the M band than young females, with a small to medium effect size (see Appendix A). Overall, our results demonstrated minor or nonsignificant sex differences in prefrontal bCON and uCOP in all three E/N/M bands in the young adult group. Because the sample size for older males was very imbalanced (males = 4; females = 20), the gender effect in older adults will be investigated later. 

### 4.6. Discussion on Using t-Tests to Compare Young and Older Adults

In this study, measurements from older adults were obtained from one visit, whereas those from young adults were derived and averaged from five visits. These two datasets created two questions: (1) whether it was appropriate to perform *t*-tests to compare these two age groups and (2) whether an imbalanced sample size would affect the statistical results.

According to [43], an independent-samples *t*-test can be used for two datasets if they are independent and normally distributed. Ref. [8] proved that the quantified metrics of uCOP and bCON from five visits by young adults could be pooled to determine group-level means for all respective metrics. Additionally, the young and older adult groups in this study were independent and approximately normally distributed based on the central limit theorem [43]. Thus, our *t*-test-based statistical analysis should provide reliable estimates or inferences regardless of repeated measures in the young adult group. However, the two datasets were imbalanced (130 for young adults vs. 24 for older adults), which may compromise the reliability and validity of the t-test results. This is one of the limitations of this study and is addressed next.

### 4.7. Limitations of the Study and Future Work

While this study provides several new and key findings, it has several limitations. First, the two datasets from the two age groups were highly imbalanced, causing inaccurate statistical inference for the study. Second, during the first visit, it was unavoidable for some of the older adults to be anxious at different levels because of the unfamiliarity of the environment and experimental setup. Such anxiety could cause motion artifacts and mental nervousness and thus lead to extra noise during data acquisition. Third, regarding data analysis, MTM is a relatively complex algorithm that is more appropriate for EEG data analysis. It may not be necessary for bbNIRS analysis. Finally, our current data analysis could not discriminate signal contamination from the superficial layers (i.e., scalp and skull) of the human forehead.

For future improvement, for the first limitation, we can perform alternative statistical analyses, such as the boot-strapping permutation test or Welch’s *t*-test. Alternatively, we can perform regular two-sample *t*-tests between older and young adults; for the latter, data from one of the five visits will be randomly selected for analysis. For the second limitation, it will be advantageous to perform future measurements with two visits or to have a practice session for older adults to reduce anxiety. For the third limitation, the Welch method [44] can be a simpler approach for performing bbNIRS spectral analysis. Finally, it is necessary to develop and implement appropriate preprocessing algorithms for bbNIRS spectral data [45] to remove possible contamination of extracranial layers for more accurate results.

## 5. Conclusions

In this study, we hypothesized that there would be significant differences in unilateral metabolic–hemodynamic coupling and bilateral connectivity in the prefrontal cortex between young and older adults. To test this hypothesis, we performed noninvasive prefrontal measurements from 24 older adults using the same two-channel bbNIRS setup and 14-min resting-state experimental protocol as those reported in [8] from 26 young adults. After comprehensive quantification of both uCOP and bCON at all three E/N/M frequencies and careful statistical comparisons, we demonstrated that young adults had significantly stronger bilateral hemodynamic connectivity in the M band than older adults with a very large effect size, whereas older adults had significantly stronger bilateral metabolic connectivity in the M band than young adults also with a very large effect size. Furthermore, the older adults exhibited significantly stronger unilateral coupling on both prefrontal sides in all three E/N/M bands, particularly with a very large effect size in the M band (>1.9). All of these results can be interpreted by neurophysiological principles while unambiguously supporting our hypothesis. The framework reported in this paper has demonstrated that the neurophysiological metrics of prefrontal uCOP and bCON are highly age-related and may have the potential to serve as neurophysiological features of human brain health that can be translatable for future clinical applications, such as for the early detection or identification of age-associated neurological disorders including Alzheimer’s disease.

## Figures and Tables

**Figure 1 bioengineering-10-01336-f001:**
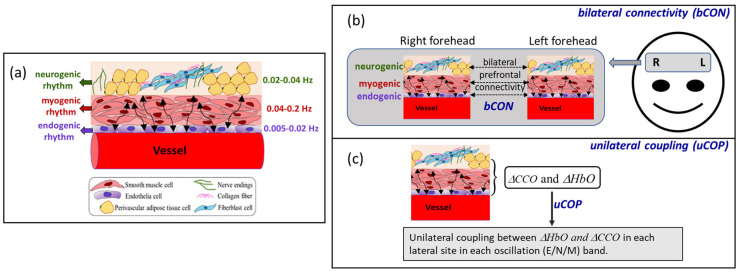
(**a**) Schematic illustration of a piece of blood vessel surrounded with three anatomical components [20,21] that facilitate spontaneous infraslow oscillations with endogenic (0.005–0.02 Hz), neurogenic (0.02–0.04 Hz), and myogenic rhythms (0.04–0.2 Hz). (**b**) Schematic illustration of the bilateral prefrontal connectivity of hemodynamic or vascular oscillation in each of the three infraslow oscillation frequencies. (**c**) Schematic illustration of unilateral prefrontal coupling between hemodynamic and metabolic oscillations in each of the three frequencies.

**Figure 2 bioengineering-10-01336-f002:**
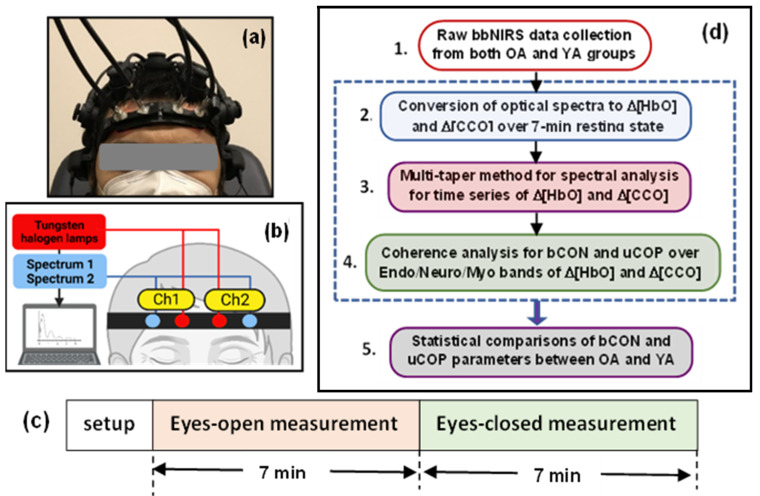
(**a**) Photograph of a human participant wearing a two-channel bbNIRS headset on their forehead along with a 19-channel EEG cap. (**b**) Schematic diagram depicting the two-channel bbNIRS setup, including two spectrometers as optical detectors, two tungsten–halogen lamps as light sources, two sets of optical fiber probes, and a control/acquisition computer. (**c**) The measurement protocol which consisted of 7-min eyes open and 7-min eyes closed measurements. It was followed for both the young and older adult studies. (**d**) Five-step flow chart for data analysis, which enables the quantification of the bilateral connectivity and unilateral coupling of mitochondrial and hemodynamic activity at rest in the three E/N/M frequency bands (see text for details). Steps 2 to 4 (outlined by the dashed box) were repeated for both age groups, enabling statistical comparisons between them.

**Figure 3 bioengineering-10-01336-f003:**
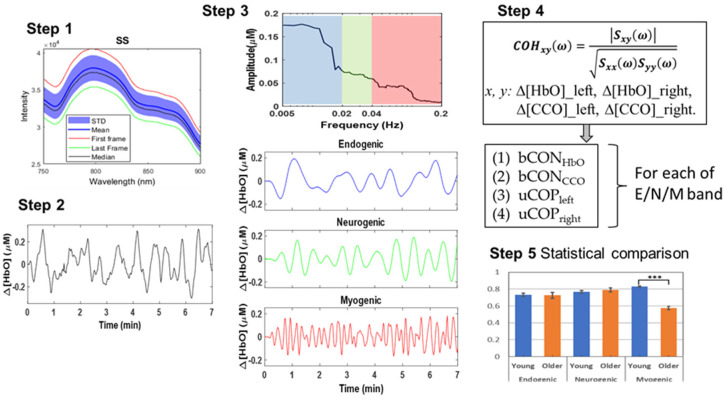
**Step 1**: An assortment of raw optical spectra acquired using bbNIRS, averaged over a 7-min collection time. The mean and median spectra are represented by shading with standard deviation and outlined by the highest and lowest spectra. **Step 2**: This panel exhibits a 7-min time series of Δ[HbO] from a random participant as an example. **Step 3**: A quantified spectral amplitude of Δ[HbO] is presented after multi-taper spectral analysis was employed. The E/N/M frequency bands are marked by blue, green, and red boxes, respectively. Accordingly, the frequency-decomposed time series of the spectral curve were obtained in respective E/N/M bands, with matched colors (blue for E, green for N, and red for M band). **Step 4**: It graphically illustrates the quantification of coherence for four pairs of Δ[HbO] and Δ[CCO] on each lateral side in each E/N/M band. **Step 5**: It graphically illustrates statistical comparisons between the two age groups for the respective parameters.

**Figure 4 bioengineering-10-01336-f004:**
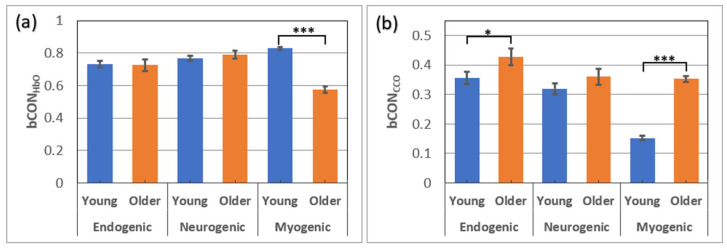
Prefrontal bilateral connectivity of (**a**) Δ[HbO] and (**b**) Δ[CCO] in older adults (*n* = 24) and young adults (*n* = 26) at rest with the eyes closed in E (0.005–0.04 Hz), N (0.04–0.02 Hz), and M (0.02–0.2 Hz) bands. The error bars represent the standard error of the mean. *: *p* < 0.05; ***: *p* < 0.001.

**Figure 5 bioengineering-10-01336-f005:**
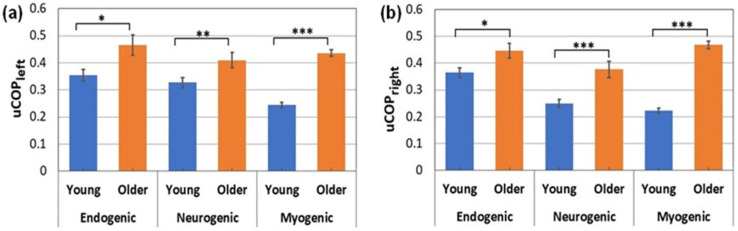
Unilateral coupling between Δ[HbO] and Δ[CCO] on the (**a**) left and (**b**) right prefrontal cortex compared between older (*n* = 24) and young adults (*n* = 26 [8]) at rest with the eyes closed in all three E/N/M bands. The error bars represent the standard error of the mean. **: p <* 0.05; ***: p <* 0.01; ****: p <* 0.001.

**Figure 6 bioengineering-10-01336-f006:**
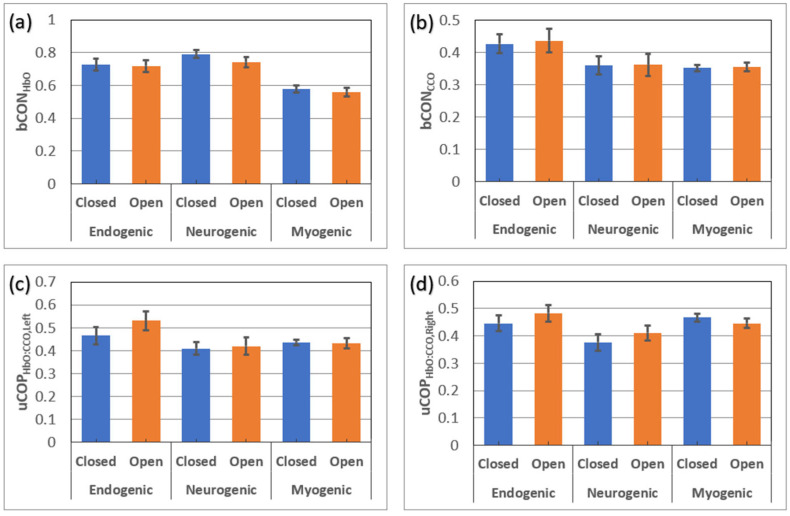
Comparisons of (**a**) bCON_HbO_, (**b**) bCON_CCO_, (**c**) uCOP_left_, and (**d**) uCOP_right_ measured from older adults under eyes-open and eyes-closed conditions. There was no significant difference in any of the metrics under the two different measurement conditions. Detailed values for each panel are available in Appendix A.

**Figure 7 bioengineering-10-01336-f007:**
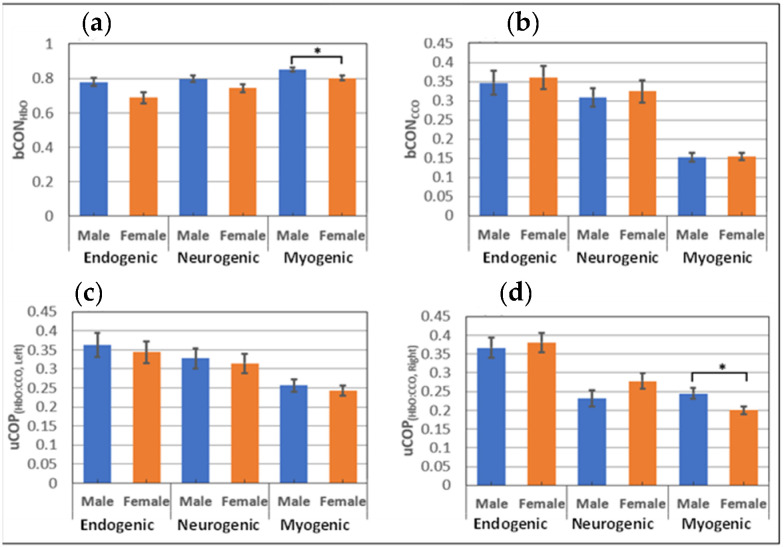
Gender comparison of prefrontal bilateral connectivity and unilateral coupling within the young adult group (males = 14; females = 12) for (**a**) bCON_HbO_, (**b**) bCON_CCO_, (**c**) uCOP_left_, and (**d**) uCOP_right_ over the E/N/M bands. The error bars are the standard error of the mean. *: *p* < 0.05. Detailed values for each panel are available in Appendix A.

**Table 1 bioengineering-10-01336-t001:** Resting-state prefrontal connectivity of (**a**) bCON_HbO_ and (**b**) bCON_CCO_. Compared between older (*n* = 24) and young (*n* = 26 [8]) adults at all three E/N/M bands.

**(a) Comparison of bCON_HbO_ between the Two Age Groups**		
**Frequency Bands**	***p* Values (*t*-Test)**	**Cohen’s d**
Endogenic	0.79	N/A
Neurogenic	0.59	N/A
Myogenic	1.1 × 10^−12^ ***	2.35
**(b) Comparison of bCON_CCO_ between the Two Age Groups**		
**Frequency Bands**	***p* Values (*t*-Test)**	**Cohen’s d**
Endogenic	0.023 *	0.37
Neurogenic	0.163	N/A
Myogenic	1.1 × 10^−21^ ***	2.99

*: It marks a significant difference between older and young adults at the significance level of *p* < 0.05. ***: It marks a significant difference between older and young adults at the significance level of *p* < 0.001.

**Table 2 bioengineering-10-01336-t002:** Unilateral couplings of the prefrontal cortex, (**a**) uCOP_left_ and (**b**) uCOP_right_, compared between older (*n* = 24) and young adults (*n* = 26 [8]) at all three E/N/M bands.

**(a) Comparison of uCOP_left_ between the two age groups**		
**Frequency Bands**	***p*-Values (*t*-Test)**	**Cohen’s d**
Endogenic	0.011 *	0.536
Neurogenic	0.0098 **	0.508
Myogenic	1.4 × 10^−18^ ***	1.96
**(b) Comparison of uCOP_right_ between the two age groups**		
**Frequency Bands**	***p*-Values (*t*-Test)**	**Cohen’s d**
Endogenic	0.023 *	0.44
Neurogenic	0.0007 ***	0.78
Myogenic	3.9 × 10^−18^ ***	2.72

*: It marks a significant difference between older and young adults at the significance level of *p* < 0.05. **: It marks a significant difference between older and young adults at the significance level of *p* < 0.01. ***: It marks a significant difference between older and young adults at the significance level of *p* < 0.001.

## Data Availability

The data presented in this study are available from the corresponding author upon request.

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
