# Peer review of "Unilateral Mitochondrial–Hemodynamic Coupling and Bilateral Connectivity in the Prefrontal Cortices of Young and Older Healthy Adults"

_bioengineering, 2023, doi:10.3390/bioengineering10111336_

Round 1
Reviewer 1 Report
Comments and Suggestions for Authors
This article by Sissons et al. used 2-channel broadband near-infrared spectroscopy (bbNIRS) approach to investigate unilateral mitochondrial-hemodynamic coupling and bilateral connectivity om humans. The primary comparison was between young adults and older adults to determine age-dependent changes in these parameters. The results focused on the resting state and some interesting differences between young adults and older adults have emerged. However, there are some problems that need to be addressed before the manuscript can be accepted for publication:
1. Excessive use of acronyms: There are too many acronyms used in the manuscript and it made the article very difficult to read. It would be preferable to reduce the acronyms down to around 6 to 7.
2. Line 73-74: Endothelial layer of vasculature does not dilate or contract. The E band is caused by endothelium-derived vasomotor activities, but the vasomotor activities are still mediated by mural cells.
3. The older adult group has 4 males and 20 females. The imbalance between sexes made the statistical analysis of gender difference highly questionable.
4. Line 141: Reference 1 does not have the details described in the text.
5. Line 186-187: The lack of EEG data reduced the overall impact of the results.
6. Line 274-277: This sentence needs to be rewritten to get the message across better.
7. Line 286-287: Data were clearly sampled differently, and it is not really a difference in sample numbers. A more comprehensive discussion regarding potential impacts of doing so is needed.
8. In the Result section, data were presented both in the figure form and the table form. This is redundant, and authors should choose one and delete the other.
9. Line 414-425: This paragraph discuss data that was not presented in the result section. It should be deleted. Alternatively, authors could move the related data from supplement to main result section.
10. Line 483: the link to supplemental data is a broken link.
Comments on the Quality of English LanguageThe clarity of many sentences in this manuscript can be improved.
Reviewer 2 Report
Comments and Suggestions for Authors
Sissons et al. indicated that there are significant differences between young and older adults (OA) in unilateral coupling (uCOP) and bilateral connectivity (bCON) in the prefrontal cortex using bbNIRS. Their work revealed that neurophysiological metrics of uCOP and bCON are highly associated with age and may have the potential to understand features for human brain health and provide an opportunity for early detection of brain abnormality. It is an interesting manuscript. There are some comments that still need to be further considered:
1. How data processing needs to further describe
2. How long was a series of Δ[HbO] and Δ[CCO] from each lateral side of the forehead of each participant that taken?
3. Cerebral hemodynamic and metabolic ISO connectivity may need to provide
4. The neurogenic component of uCOPHbO-CCO should be provided in comparison in the left prefrontal cortex and the right prefrontal cortex
5. Since the work was aimed to clinical application, it should include some disease-related patients
Round 2
Reviewer 2 Report
Comments and Suggestions for Authors
Thanks to the authors for their thorough response to these comments. I believe you’ve done a good job of responding to those comments, and with additional changes, I’ll be happy to accept this paper.
Author Response
Thank this reviewer for his or her positive feedback for our revision. There was no given point for further revision.